# The antidepressant drug vilazodone is an allosteric inhibitor of the serotonin transporter

Per Plenge[1,6], Dongxue Yang[2,6], Kristine Salomon [1], Louise Laursen [1], Iris E. Kalenderoglou [1], Amy H. Newman [3], Eric Gouaux [2,4], Jonathan A. Coleman [2,5✉] & Claus J. Loland [1✉]

Depression is a common mental disorder. The standard medical treatment is the selective serotonin reuptake inhibitors (SSRIs). All characterized SSRIs are competitive inhibitors of the serotonin transporter (SERT). A non-competitive inhibitor may produce a more favorable therapeutic profile. Vilazodone is an antidepressant with limited information on its molecular interactions with SERT. Here we use molecular pharmacology and cryo-EM structural elucidation to characterize vilazodone binding to SERT. We find that it exhibits non-competitive inhibition of serotonin uptake and impedes dissociation of [3H]imipramine at low nanomolar concentrations. Our SERT structure with bound imipramine and vilazodone reveals a unique binding pocket for vilazodone, expanding the boundaries of the extracellular vestibule. Characterization of the binding site is substantiated with molecular dynamics simulations and systematic mutagenesis of interacting residues resulting in decreased vilazodone binding to the allosteric site. Our findings underline the versatility of SERT allosteric ligands and describe the unique binding characteristics of vilazodone.

[1] Laboratory for Membrane Protein Dynamics. Department of Neuroscience, Faculty of Health and Medical Sciences, University of Copenhagen, Copenhagen, Denmark. [2] Vollum Institute, Oregon Health & Science University, Portland, OR, USA. [3] Medicinal Chemistry Section, Molecular Targets and Medications Discovery Branch, National Institute on Drug Abuse - Intramural Research Program, National Institutes of Health, Baltimore, MD, USA. [4] Howard Hughes Medical Institute, Oregon Health & Science University, Portland, OR, USA. [5] Present address: Department of Structural Biology, University of Pittsburgh, Pittsburgh, PA, USA. [6] These authors contributed equally: Per Plenge, Dongxue Yang. ✉email: coleman1@pitt.edu; cllo@sund.ku.dk

Synaptic transmission is a fundamental process which underlies neuronal communication[1]. Neurotransmitters within synaptic vesicles are released from the pre-synaptic neuron and diffuse across the synapse, binding to post-synaptic receptors and thereby mediate downstream signaling[2,3]. Serotonin is a neurotransmitter which is involved in modulating many key brain functions including sleep, cognition, mood, appetite, and sex drive as well as other important physiological processes such as gastrointestinal mobility and blood clotting[4]. The serotonin transporter (SERT) removes serotonin from synapses, recycling it into the pre-synaptic neuron and terminating signaling[5–7]. SERT belongs to the family of neurotransmitter transporters known as neurotransmitter sodium symporters (NSSs) which also includes the dopamine, norepinephrine, GABA, and glycine transporters as well as numerous bacterial homologs[8]. NSSs use the energy stored in the sodium gradient to drive the thermodynamically "uphill" transport of the substrate. NSSs harbor a conserved 3D fold which consists of a 10 transmembrane (TM) core region arranged into two inverted-topological repeats, known as the LeuT fold[7,9–16], as well as one or two additional C-terminal TMs. NSSs are proposed to utilize an alternating access mechanism to transport substrate[17] where a rocking bundle of TM1, 2, 6, and 7 has been suggested to control inward versus outward accessibility to the central substrate-binding site relative to a stable scaffold domain of TM3, 4, 8, and 9[14,18,19]. It is hypothesized that in the transporter's apo form, the conformational equilibrium will be biased towards an outward-open conformation, primed to bind ligands, such as neurotransmitter and ions ($Na^+$, $Cl^-$). Upon binding, the protein will rearrange to an occluded conformation, followed by opening to the cytosol where the substrate and ions will be released. The transporter then transitions back to the outward-open conformation[6,20,21]. For SERT and LeuT, the return step can involve a counter-transport of $K^+$[22,23].

SERT is the target of therapeutics used to treat major depressive, anxiety, obsessive-compulsive, eating, and post-traumatic stress disorders[6,24,25]. Imipramine (IMI, Supplementary Fig. 1) was the first tricyclic antidepressant (TCA) to be medically used and was found to block both SERT and the norepinephrine transporter (NET)[26]. However, IMI was also found to exhibit a large number of side effects including neurological, pulmonary, and gastrointestinal complications, as well as toxicity[27]. Selective serotonin reuptake inhibitors (SSRIs) were developed to bind SERT with high affinity and specificity, resulting in better tolerance and fewer side effects[28,29]. However, many individuals who are prescribed SSRIs also still report a myriad of side effects including sexual dysfunction, weight loss or gain, anxiety, nausea, headaches, dizziness, insomnia, and dry mouth, many of which pose barriers to adherence[30]. All investigated drugs have been shown to exert their inhibitory effect as competitive inhibitors by binding to the central binding site for serotonin, located about halfway across the membrane, also known as the S1 site[31–33].

NSSs possess an extracellular vestibule, serving as an entryway to the S1 site, which also harbors a potential allosteric binding site for ligands, deemed the S2 site[31,32,34–36]. Drugs that target allosteric sites can be superior to competitive drugs that target orthosteric sites because the allosteric sites are typically less conserved and can therefore demonstrate higher drug target selectivity[37,38], resulting in fewer side effects. Drugs that act at allosteric sites can also provide a different therapeutic indication because they can act as more than an on/off switch and can function as activity-modulators (e.g., benzodiazepines) or possess activity only in specific body regions due to environmental selectivity (e.g., acetaminophen/paracetamol). Accordingly, the allosteric site in SERT, as well as in other NSS proteins, is an attractive target that has recently gained attention[39–43].

Several drugs bind to the S2 site in SERT, including S-citalopram (S-CIT, Supplementary Fig. 1)[34,44] and clomipramine[34]. However, so far all therapeutic drugs that possess an allosteric component bind to the S2 site with low affinity, thus rendering their allosteric binding therapeutically irrelevant. For example, S-CIT binds to the S1 site of SERT with an affinity of 5 nM and to the S2 site with 5 μM[34,45]. We have reported on the first small-molecule inhibitor possessing high affinity to the S2 site, Lu AF60097 (Supplementary Fig. 1), that inhibits 5-HT transport by SERT via a mixed competitive and non-competitive mechanism, suggesting it binds both the allosteric and central sites[40]. Lu AF60097 has an allosteric potency of ~30 nM when [$^3$H]IMI was bound to S1. The allosteric interaction has a synergistic effect, which could alleviate side effects from IMI treatment while preserving its therapeutic effects.

Vilazodone (viibryd®; VLZ, Supplementary Fig. 1) is a clinically approved SSRI which also has activity as a 5-HT$_{1A}$ receptor partial agonist[46]. Its action on the 5-HT$_{1A}$ receptor is thought to provide for its faster onset of action relative to the classical SSRIs[47], as has been observed for vortioxetine[48]. The common side effects of VLZ include nausea, diarrhea, and headaches, which are also common for other SSRIs. In contrast, VLZ is not as frequently associated with a reduction in sexual dysfunction[49–51], the most common cause of discontinuation of SSRI treatment[52]. Also, weight gain is reduced with VLZ treatment relative to other SSRIs[49–51]. However, despite these intriguing properties, the molecular basis for VLZ binding to SERT remains an open question. The only report addressing the VLZ binding site is a computational approach using ligand docking and molecular dynamics (MD) simulations[53]. In this study, VLZ binds to the S1 site with the main interacting residues overlapping with those that bind other TCAs and SSRIs. However, due to the long and potentially bitopic nature of the molecule, its 5-cyanoindole group protrudes into the S2 site and disrupts an extracellular salt bridge between Arg104 in TM1 and Glu493 in TM10.

Here we use molecular pharmacology and cryo-EM structural analysis to elucidate the SERT:VLZ interaction. We find that VLZ exhibits non-competitive inhibition of serotonin transport. We show that VLZ impedes the dissociation of S1-bound [$^3$H]IMI with low nanomolar affinity suggesting a strong association to the S2 site without involving residues in the S1 site. This is supported by a cryo-EM structure of the SERT:IMI:VLZ complex, which reveals that VLZ binds to the allosteric (S2) site that includes an aromatic pocket formed by TM10, 11, and 12. Accordingly, our studies expand the three-dimension volume of the allosteric site of SERT and demonstrate a mode of action of the clinically approved antidepressant drug VLZ, which could account for its distinct therapeutic profile relative to other SSRIs.

## Results

**The VLZ binding site is distinct from other antidepressants.** VLZ is one of the newer SSRIs on the market. It was approved for the treatment of major depressive disorders in the US in 2011 and reached more than two million prescriptions per year by 2015[54]. Despite the therapeutic success, understanding its mechanistic underpinnings at SERT is limited. To examine the SERT:VLZ interaction in detail, we first measured its ability to inhibit [$^3$H]5-HT transport. COS-7 cells were transiently transfected with SERT WT and preincubated with increasing concentrations of VLZ to obtain binding equilibrium before adding [$^3$H]5-HT (Fig. 1a). We found that VLZ inhibits [$^3$H]5-HT transport with an apparent affinity ($K_i$) of 1.1 nM (Supplementary Table 1). This is in accordance with previously reported data[55] and approximately 5 times as potent as S-CIT, which has an apparent affinity ($K_i$) of

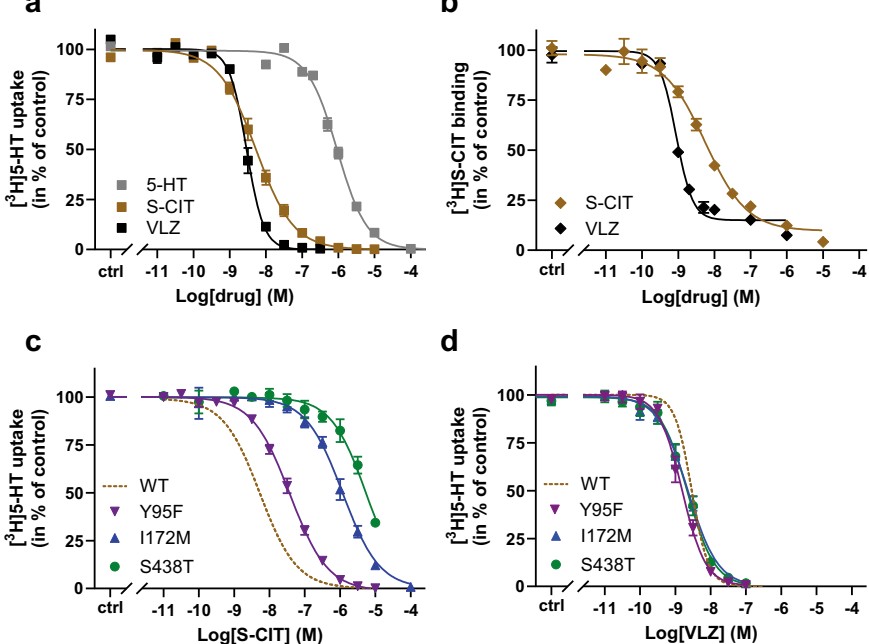

**Fig. 1 Characterization of VLZ binding to SERT WT and S1 mutants. a** Inhibition potency of VLZ (black) and S-CIT (brown) compared to 5-HT (gray). The affinity for VLZ is 5-fold higher than for S-CIT in inhibiting [³H]5-HT transport with $K_i$ values of 1.06 [0.90; 1.25] and 5.21 [4.19; 6.48] nM, respectively (n = 6). **b** Equilibrium binding inhibition for S-CIT and VLZ displacement of [³H]S-CIT. The $K_i$ for VLZ is 7-fold higher than for S-CIT (0.68 [0.56; 0.83] and 5.13 [2.85; 3.76] nM, respectively). **c, d** The affinity for S-CIT but not VLZ is affected by mutations in the SERT S1 site. S-CIT binding is decreased by 7- and 192- and 625-fold by the mutations Y95F (purple), I172M (blue) and S438T (green), respectively relative to WT (dotted line of data in **a**). The mutations do not significantly affect the affinity for VLZ. $V_{MAX}$, $K_M$, $K_i$, and n-values for 5-HT, S-CIT, and VLZ binding to SERT WT and S1 mutants are shown in the Supplementary Table 1. Experiments are performed in triplicates on intact COS7 cells transiently expressing SERT WT or mutants. Data are shown as means ± S.E. (error bars), n = 4–17. Source data are provided as a Source Data file.

5.2 nM (Supplementary Table 1). In equilibrium binding experiments using [³H]S-CIT (Fig. 1b), VLZ showed a similar affinity with a $K_i$ of 0.7 nM, which is also about 5-times the affinity of S-CIT. All SSRIs and TCAs inhibiting SERT bind with high affinity to the S1 site[56–62] and a recent MD simulation also suggested that VLZ binds at this site[53]. S1-site drug binding is dependent on the side-chain interactions with at least two out of three key residues: Tyr95, Ile172, and Ser438 and the MD simulation of VLZ binding also suggested that all three of these residues are involved in its binding pose. We therefore wished to test this experimentally by measuring the impact of mutagenesis of these residues on VLZ affinity. Using the same experimental setup as in Fig. 1a, we found that all three mutants are capable of transporting 5-HT with a $K_M$ similar or slightly higher than observed for SERT WT (Supplementary Fig. 2, Supplementary Table 1). The apparent affinity for S-CIT was decreased 7, 190, and 625-fold for Y95F, I172M, and S438A, respectively (Fig. 1c, Supplementary Table 1), in line with previous observations[59]. Surprisingly, we did not find any effect of the mutations on the apparent affinity for VLZ (Fig. 1d, Supplementary Table 1), suggesting that the drug binds in a different site than the other tested antidepressants.

**SERT possesses a high-affinity allosteric binding site for VLZ.** Our results from the S1 site mutants raised the possibility that VLZ may bind to a different site in SERT. A classical approach to determine whether VLZ binds competitively or non-competitively to 5-HT is to perform a [³H]5-HT uptake experiment in the presence of increasing VLZ concentrations (Fig. 2a). The experiment showed that VLZ causes a decrease in the $V_{MAX}$ of [³H]5-HT without having any significant effect on $K_M$ (Supplementary Table 2). This is in accordance with non-competitive

binding between 5-HT and VLZ suggesting that VLZ may not bind to the central S1 site. We therefore investigated VLZ binding to the S2 site. To investigate S2 binding, we and others have previously shown that binding of SERT inhibitors to the S2 site can be characterized by their ability to impede the dissociation of an S1-bound radioligand[34,40,63,64]. Thus, we prepared membranes of COS-7 cells transiently expressing SERT WT and preincubated them with [³H]IMI or [³H]S-CIT, followed by the addition of VLZ. We then monitored the effect of VLZ on the dissociation rate of the radiolabeled drug. Indeed, over a range of concentrations, VLZ dose-dependently slowed the rate of [³H]IMI unbinding (Fig. 2b). The results show that VLZ inhibits [³H]IMI dissociation with an allosteric potency of 14 nM (Fig. 2c, Table 1). This is several orders of magnitude more potent than previously reported for any other drug[34,63]. VLZ also inhibits [³H]S-CIT dissociation, albeit with a lower potency of 250 nM (Fig. 2c). The high allosteric potency of VLZ could suggest that it binds in the extracellular vestibule and thereby slows dissociation by occlusion of the exit pathway from the S1 site.

**Cryo-EM structures of SERT-15B8-Fab:IMI:VLZ complex.** To investigate the VLZ binding location and conformation, we determined the structure of the SERT:IMI:VLZ complex (PDB: 7LWD). We purified human SERT using an N- and C-terminally truncated construct (ΔN72/C13) in the presence of VLZ and IMI. The 15B8-Fab was used to guide particle alignment for single-particle cryo-EM analysis. We have previously shown that the binding of this Fab to SERT does not perturb SERT transport activity[20]. Here we further investigated whether Fab binding to SERT affects the binding of VLZ. The results show that the VLZ dissociation rate is unaffected by the presence of Fab, irrespective of the S1-bound radioligand, suggesting that the Fab fragment

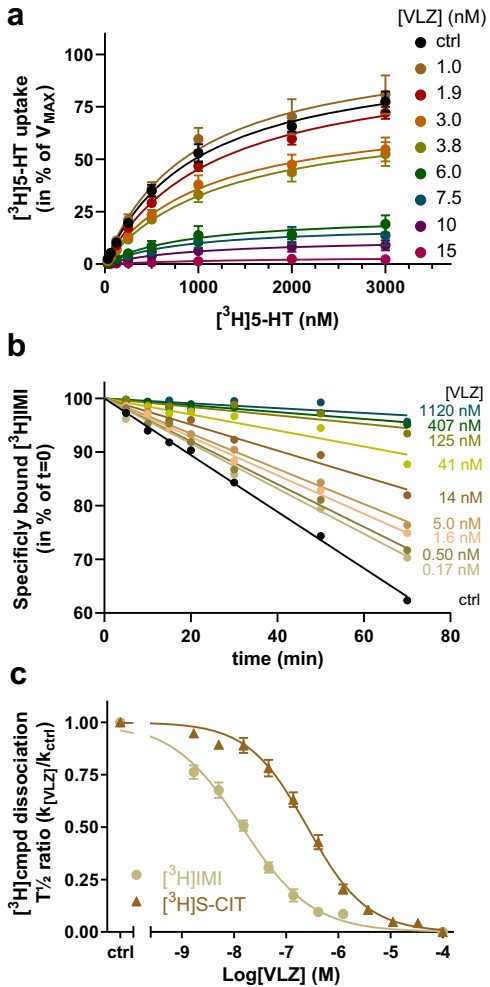

**Fig. 2 Indications of VLZ as an allosteric inhibitor of SERT WT.**
**a** Saturation uptake experiments for [³H]5-HT transport as a function of increasing VLZ concentrations (0–15 nM) are compatible with a non-competitive inhibition for VLZ. Colors of lines and symbols represent various concentrations of VLZ defined in legend. The $V_{MAX}$ for [³H]5-HT decreases with increasing VLZ concentration. Data are shown in percent of 5-HT $V_{MAX}$ in the absence of VLZ ($V_{MAX}$ in the absence of VLZ is 4899 ± 312 cpm/min/10⁵ cells). The $K_M$-value for 5-HT is largely unaffected by increasing VLZ concentrations (Supplementary Table 2). Non-specific [³H]5-HT binding was assessed in the presence of 1 μM paroxetine. Experiments are performed in triplicates on intact COS-7 cells transiently expressing SERT, $n = 3$–8. **b** Representative experiment for the assessment of VLZ allosteric potency determined by its influence on [³H] IMI dissociation. The slope for [³H]IMI dissociation ranges from 0.528 ± 0.012 min⁻¹ in the absence of VLZ (ctrl, black line) to 0.045 ± 0.019 min⁻¹ in the presence of 1120 nM VLZ (blue line). Data are converted to the dose-response curve in (**b**) to determine the allosteric potency (see Methods). **c** Allosteric potency of VLZ determined as concentration-dependent inhibition of [³H]IMI and [³H]S-CIT dissociation. The radioligands were prebound to SERT WT prior to the addition of VLZ in the indicated concentrations. Data are plotted as the effect of VLZ on the [³H]IMI and [³H]S-CIT dissociation rate ($k_{[VLZ]}$) relative to the rate in the absence of VLZ ($k_{[ctrl]}$). The allosteric potency of VLZ is ~17-fold higher in the presence of prebound [³H]IMI (circles, tan) relative to [³H]S-CIT (triangles, brown), with IC₅₀ = 14.1 [11.8; 16.8] and 250 [203; 307] nM, respectively (mean [S.E. interval]), $n = 10$ ([³H]IMI) and 7 ([³H]S-CIT). Data are shown as means ± S.E. (error bars). [³H]cmpd: [³H]compound (being either [³H]IMI or [³H]S-CIT). Source data are provided as a Source Data file.

does not influence VLZ binding to SERT (Supplementary Fig. 3). A density map of the SERT:15B8-Fab:IMI:VLZ complex was obtained at overall resolution of 3.65 Å with the highest local resolution extending to 2.9 Å in the core TMs near the drug binding sites (Fig. 3a, Supplementary Fig. 4). The SERT and drug densities were of good quality for most key functional regions with the side chains of most large residues well-fitted into their corresponding density features.

The overall structure is similar to the X-ray structure of SERT complexed with S-CIT (PDB: 5I73) and paroxetine (PDB:5I6X)[44] and the transporter has been stabilized in an outward-open conformation with an intact Arg104-Glu493 salt bridge (Fig. 3b). IMI exhibits a binding pose similar to the one found previously by MD simualtions[40] and also aligns with S-CIT and paroxetine binding in the S1 site[44] (Fig. 3e). The chemically equivalent amine group of IMI occupies subsite A as it does for S-CIT, while the dibenzazepine group localizes in subsites B and C (Fig. 3e).

In the allosteric site, about 12 Å above the S1-bound IMI, the bulky backbone of VLZ adopts a nearly linear binding pose across the extracellular vestibule. The heteroaryl piperazine moiety of VLZ overlaps with S2-bound S-CIT and Lu AF60097 (Supplementary Fig. 1), hindering the entry pathway to the S1 site and ensuring the blockade of IMI dissociation from SERT. VLZ binding is unlike the other allosteric site binding inhibitors, S-CIT and Lu AF60097, as these adopt a triangular pose with the cyano-group extending toward the non-helical region of TM6. The quinolinone moiety of Lu AF60097 protrudes into a subsite near the tip of the extracellular loop (EL) 4 (Fig. 3f). In contrast, the indole ring of VLZ is nestled in a hydrophobic cavity constituted by residues from TM10, TM11, TM12, and EL6 (Fig. 3f), allowing for an almost linear pose of the drug. The extension of the indole ring of VLZ, close to TM12, capped by EL6 and flanked by Tyr579, indicates the promiscuity of the allosteric site in NSSs. The diversity of chemical structures of allosteric inhibitors results in different binding modes in the S2 site, which may result in distinct pharmacological profiles[65].

We examined detailed views of the SERT residues forming the ligand-binding sites (Fig. 4), in the S1 site. The tertiary amine group of IMI interacts with the carboxylate of Asp98 and forms cation-π interaction with Tyr95 from TM1 (Fig. 4b). The dibenzazepine group is encompassed by Tyr176, Phe341, and Thr497 and interacting with Ile172 and Tyr175 through hydrophobic and aromatic interactions (Fig. 4b). In the allosteric site, Tyr495, Pro499, Ser559, Pro560, Pro561, and Tyr579 form extensive hydrophobic interactions with the indole on VLZ (Fig. 4c). Glu494 likely forms a salt bridge with the protonatable piperazine nitrogen, whereas the piperazine ring system forms a hydrophobic interaction with Phe556 (Fig. 4c). The benzofuran ring extends close to the central site allowing the formamide group to interact with IMI (3.8 Å to the tertiary amine group) and with Gln332 in TM6a. Glu493 and Arg104 form a salt bridge "above" the benzofuran moiety while Phe335, from "underneath", participates in aromatic interactions (Fig. 4c).

**MD simulation of VLZ binding reveals two possible poses**. We calculated the local resolution of IMI and VLZ in our reconstructions to 2.9 Å and 3.4 Å, respectively (FSC, 0.143). Their modeling into their respective sites of the cryo-EM structure resulted in correlation coefficients (CC) of 0.81 and 0.79. However, both molecules are pseudosymmetric. Thus, the possibility of the ligands binding in a pose rotated by 180° rather than the hypothesized one, was not ruled out. When the ligands—in this "flipped" orientation—were fit into the density features, both had a CC of 0.69 after refinement (Supplementary Fig. 5), supporting the hypothesized pose. To investigave this further, we performed

| Table 1 Transport and binding kinetics for SERT WT and S2 site mutants. | | | | | |
|---|---|---|---|---|---|
| | [$^3$H]5-HT transport | | | Allosteric potency [$^3$H]IMI dissoc. | |
| | 5-HT $V_{MAX}$ (pmol/min/10$^5$ cells) | 5-HT $K_M$ (nM) | n | VLZ IC$_{50}$ (nM) | n |
| WT | 26270 ± 1967 | 873 [781; 976] | 17 | 14.1 [11.8; 16.8] | 10 |
| R104K | 1586 ± 448 | 693 [577; 834] | 5 | 296 [248; 353]$^a$ | 5 |
| Q332A | 1151 ± 64 | 3070 [2710; 3490] | 4 | 218 [181; 264]$^a$ | 3 |
| F335A | 2368 ± 394 | 523 [506; 540] | 3 | 6074 [5310; 6949]$^a$ | 4 |
| E493N | 12764 ± 2950 | 699 [556; 878] | 3 | 184 [175; 193]$^a$ | 3 |
| E494Q | 26152 ± 4068 | 1100 [950; 1270] | 3 | 443 [399; 491]$^a$ | 3 |
| Y495A | 1553 ± 369 | 4220 [2840; 6280] | 5 | 3383 [2882; 3971]$^a$ | 3 |
| F556A | 11062 ± 1509 | 482 [418; 555] | 5 | 1085 [929; 1267]$^a$ | 10 |
| P561G | 49775 ± 7634 | 1150 [1030; 1280] | 3 | 93 [74; 118]$^a$ | 5 |
| Y579A | 49685 ± 6264 | 1540 [1350; 1760] | 3 | 54 [37; 80]$^a$ | 4 |

5-HT transport is performed on intact COS7 cells transiently expressing SERT WT or mutants. $V_{MAX}$, and $K_M$ are calculated based on IC$_{50}$ and catalytic activity as described in "Methods" section. Cells are incubated with [$^3$H]5-HT for three minutes. VLZ is added 30 min prior to [$^3$H]5-HT to obtain equilibrium. Values are calculated from non-linear regression analysis of [$^3$H]5-HT uptake and [$^3$H]IMI dissociation data. Data are shown as mean and either ± S.E. (for $V_{MAX}$) or [S.E. interval], the latter calculated from pIC$_{50}$ ± S.E. Uptake experiments are performed in triplicates. Allosteric potency is calculated from a dose-dependent inhibition of [$^3$H]IMI dissociation from SERT (see Methods).
$^a$Significantly different from allosteric potency for SERT WT ($P < 0.0001$, Ordinary one-way ANOVA with Tukey's multiple comparisons test).

200 ns of all atom molecular dynamics (MD) simulations of the final refined SERT structures with IMI and VLZ bound at their binding sites, either in the hypothesized or flipped pose (Fig. 5). Both simulations converged after ~50 ns to root mean square deviations (RMSDs) of $2.2 ± 0.1$ Å and $2.0 ± 0.2$ Å (mean ± S.E.), for the hypothesized and flipped poses, respectively (Supplementary Fig. 6, Supplementary Movie 1 and Supplementary Movie 2). The hypothesized and the flipped poses after MD simulation were used to fit back into the electron density. After real space refinement, the flipped pose for IMI came outside the density and produced several clashes with atoms within the ligand itself. For VLZ, both poses showed low fluctuation in the MD simulations with an RMSD of the ligand's heavy atoms between the initial and final simulated structures, of 1.5 Å and 1.7 Å for the hypothesized and flipped poses, respectively (Fig. 5a, b). Both poses were also fit back into the electron density without major clashes (Fig. 5c–f). The molecular mechanics Poisson–Boltzmann surface area (MMPBSA) method was used to calculate the protein-ligand binding free energy of each VLZ pose. The free energy of binding ($\Delta G_{bind}$) was not significantly different between the hypothesized and flipped poses with $\Delta G_{bind} = -10 ± 3$ kcal/mol and $-14 ± 3$ kcal/mol, respectively. This opens the possibility that VLZ may bind with equally high affinity in both poses. However, when we viewed both fits of VLZ from the plane of the membrane (Fig. 5e, f), the indole ring in the hypothesized pose complies better with the electron density. Taken together this suggests that VLZ can bind in both poses, but the hypothesized pose is predominant in our cryo-EM structure.

**Mutation of VLZ binding residues decreases allosteric potency.** Based on our cryo-EM structure and MD simulations, we define the predominant allosteric binding site for VLZ as the hypothesized one. It is composed of residues, Arg104 (TM1), Gln332 and Phe335 (TM6), Glu493, Glu494, and Tyr495 (TM10), Phe556 (TM11), Ser559 and Pro561 (EL6) and Tyr579 (TM12) (Fig. 6a). All residues have hydrophobic interactions with VLZ. To further investigate the involvement of each side chain, we mutated each residue, one-by-one, and monitored the impact on the allosteric potency of VLZ (Fig. 6b, Table 1). All mutants possessed detectable 5-HT transport activity and most mutants had a $K_M$ for 5-HT transport with less than a two-fold deviation from SERT WT. The two exceptions were Q332A and Y495A, which showed a 3.5- and 4.8-fold increase in $K_M$, respectively. The decreased apparent affinity coincided with a marked drop in transport velocity, suggesting that the two mutations may change the

conformational equilibrium of SERT. All of the mutants showed a decreased allosteric potency relative to SERT WT, consistent with an impairment of VLZ binding. The decrease in allosteric potency ranged from ~4-fold for Y579A to 430-fold for the F335A mutant (Table 1). The mutants fall into three categories with respect to their effect: F579A and P561Q exhibit less than 7-fold decrease in allosteric potency. An intermediary group of R104K, Q332A, E493N, and E494Q show a 13- to 30-fold decrease in potency, whereas F335A, T495A, and F556A have 77-, 240- and 430-fold decrease, respectively. The fold change in allosteric potency is not only dependent on the interaction of the residue and VLZ, but also on the ability of the substituted residue to compensate for the WT environment. The isolated contribution of each residue might deviate somewhat from the observed affinity changes. Taken together, we find that all mutants investigated produced a significant decrease in the allosteric potency for VLZ, supporting the binding location of the cryo-EM structure.

## Discussion

Depression is a major worldwide problem. In the US population, above the age of 18 the prevalence was 7.8% in 2019[66] and with the entry of the COVID-19 pandemic, this is expected to rise further. More than a third of patients suffering from the major depressive disorder are non-responsive to classic antidepressant treatment[67], supporting the value of a larger selection of drugs with a variety of pharmacological actions in identifying the optimal treatment regimen for each individual. So far, all antidepressant drugs targeting SERT have been reported to bind with high affinity to the orthosteric, central binding site. Here we report that VLZ binds with high affinity to the S2 site located to the SERT extracellular vestibule. The finding is substantiated by pharmacological and structural data showing (i) that VLZ binds non-competitively to 5-HT (Fig. 2a), (ii) its affinity is unaffected by central S1 mutations (Fig. 1d), (iii) it has a profound effect on [$^3$H]IMI dissociation (Fig. 2c), which is decreased by mutation of its binding site (Fig. 6b), and (iv) high-resolution structural evidence supported by MD simulation for the association of VLZ to the S2 site (Figs. 3–5). With an allosteric potency of ~14 nM, VLZ binds with higher S2 affinity than any other molecule reported. This is supported by its linear binding pose spanning approximately 19 Å, wedging between TM1 and 6 and capped by Tyr579 in TM12. Moreover, the VLZ binding site is different from the allosteric binding site for S-CIT[44]. Though there are overlapping regions in the binding sites, S-CIT is a smaller molecule and kinks "downwards" with its fluoro-group between TM10 and 11. The

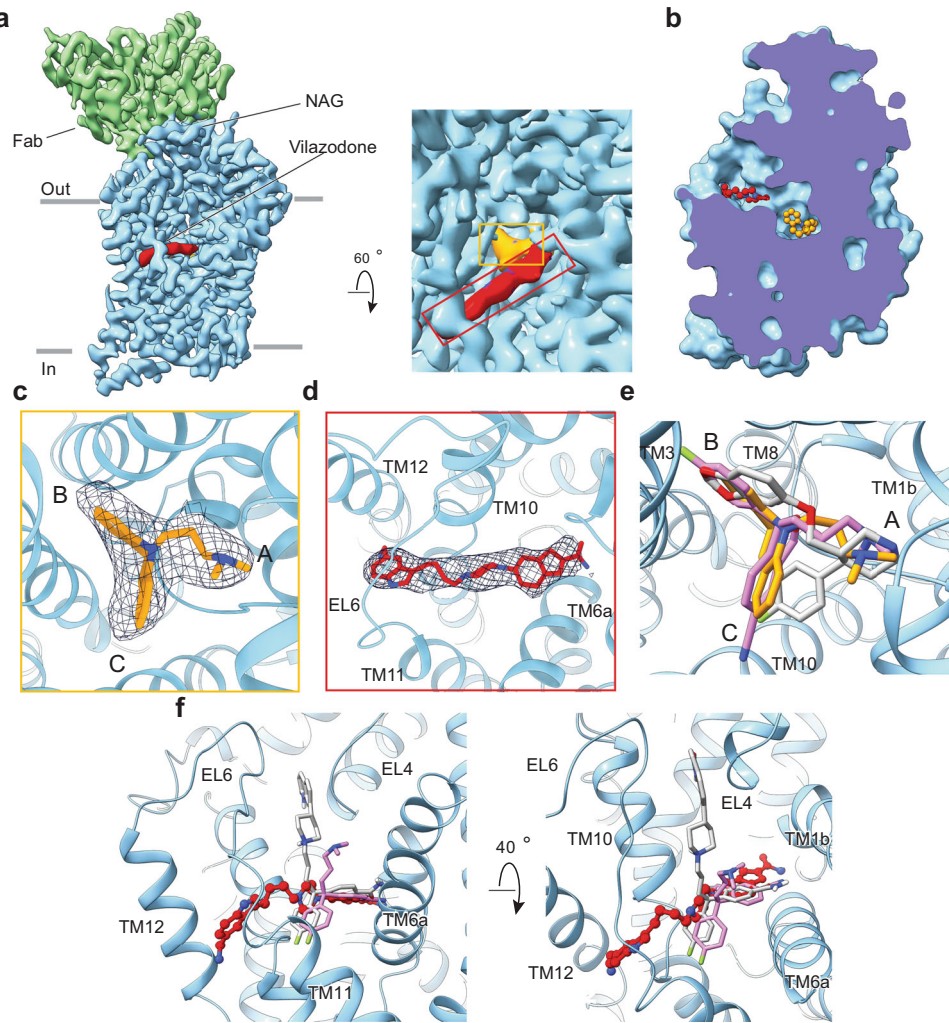

**Fig. 3 Cryo-EM structure of SERT:IMI:VLZ complex. a** The overall reconstruction of the SERT:15B8-Fab:IMI:VLZ complex at 3.65 Å (PDB: 7LWD). Right panel shows the zoomed allosteric and central sites. SERT is colored in light blue and the 15B8 Fab is shown in green. IMI and VLZ densities are shown in yellow and red, respectively. **b** Surface representation of the outward-open structure of SERT, viewed parallel to the membrane. The open extracellular pathway and the closed intracellular pathway are displayed. IMI and VLZ are shown as ball sticks. **c** Zoomed view of the IMI (yellow) binding site. IMI is shown fit into a density feature at the S1 site with subsite A, B, and C and the involved SERT backbone structures (light blue) annotated. Parts of SERT have been removed for clarity. **d** VLZ (red) is shown fit into a density feature at the allosteric site with the central SERT backbone structures annotated. Parts of SERT have been removed for clarity. **e** Comparison of the IMI (yellow) binding pose with the binding poses for S-CIT (light magenta, PDB code:5I73) and paroxetine (gray, PDB code:5I6X) within the S1 site. The main chain position of SERT from the IMI:VLZ complex is shown in light blue. **f** Comparison of VLZ (red) with S-CIT (light magenta, PDB code:5I73) and Lu AF60097 (gray). The main chain position of SERT from the IMI:VLZ complex is shown in light blue.

binding site is also markedly different from the experimental compound Lu AF60097, which has similarities to the VLZ structure as well as to S-CIT. Lu AF60097 binds similarly to S-CIT but has been suggested to create a ~90° kink "upwards" with its N-substituted ring thus occluding the presumed exit pathway by interacting with TM10[40]. In contrast, VLZ twists around TM10 and protrudes all the way to TM12. This novel binding mode expands the size of the S2 site and highlights the versatility of ligands that could bind at this site (Fig. 3f).

IMI was first discovered in the 1950s and marked a new era of molecular pharmacology which revolutionized the treatment of psychiatric disorders. This not only gave clinicians better tools for treating mood disorders but also directly enabled advancements in our understanding of neurotransmission and neurotransmitter reuptake through the seminal work of Arvid Carlsson and Julius Axelrod[68,69]. Our cryo-EM structure provides a high-resolution view of IMI binding to SERT now more than 60 years later. We find that compared to the binding pose of other TCAs[56] and

SSRIs[70], the binding of IMI shows a similar association to the S1 site with an equal distribution of drug moieties in subsites A, B, and C (Fig. 3d).

A recent MD simulation[53] has suggested a binding pose for VLZ to the S1 site with a protrusion into the S2 site. Based on our cryo-EM structure, we cannot rule out that VLZ would associate with the S1 site in the absence of IMI. However, if the suggested pose was predominant, we would expect our mutagenesis study of the S1 site to show a decrease in VLZ affinity (Fig. 1d). We would also expect our [3H]5-HT saturation uptake experiments to show, at least in part, a competitive inhibition by VLZ (Fig. 2a). Another possible explanation is that VLZ has two binding sites of similar affinity. We note that the Hill slope of ~2 for VLZ inhibition of [3H]5-HT uptake (Fig. 1a) opens this possibility.

The solved SERT structure is similar to previously solved structures with antidepressants bound in the S1 site[44,70,71] though the Arg104-Asp493 salt bridge is intact the S1-bound ligands are solvent accessible to the extracellular environment. Accordingly,

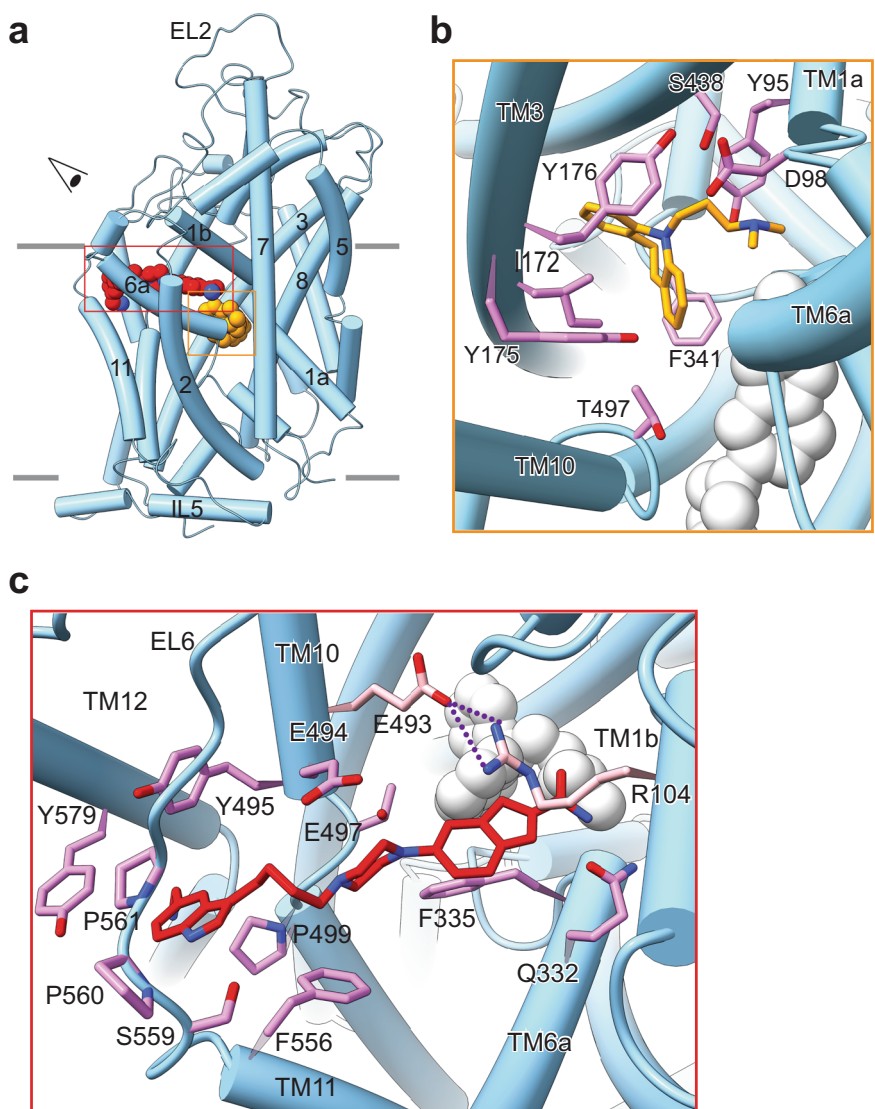

**Fig. 4 Detailed representation of the binding sites for IMI and VLZ in SERT. a** Overall view of SERT:IMI:VLZ complex in cartoon representation; IMI (yellow) and VLZ (red) are depicted as spheres. The "eye" represents the angle view depicted in (**b**) and (**c**). **b** Detailed representation of the IMI binding site in S1. IMI binding is mainly formed by side chain residues in TM1, 3, 6, 8, and 10. Residues that interact with IMI are annotated and shown in violet sticks. IMI is shown in yellow sticks and VLZ as gray sphere. **c** Detailed representation of the VLZ binding site in S2. Residues that interact with VLZ are annotated and shown in violet sticks, Arg104 and Glu493 are shown in pink. VLZ is shown in red sticks and IMI as gray sphere. The salt bridge formed between Arg104 and Glu493 is shown as purple dashed lines. TM: Transmembrane, EL: Extracellular loop, IL: Intracellullar loop.

the conformation is outward-open. This is in contrast to the solved LeuT structures where an intact salt bridge excludes solvent to the S1 site and is thus an outward-occluded conformation. It is indeed possible that the SERT transport cycle also includes an outward-open conformation with a broken salt bridge.

Could there be a connection between the VLZ binding pose and its reduction in adverse effects relative to other SSRIs? A linkage between binding pose and therapeutic effect is a well-known phenomenon for biased agonism in G-protein coupled receptors, especially for bitopic ligands[72,73], similar to VLZ, but correlations have also been reported in transporters. The stimulant effect of cocaine is due to its inhibition of the DAT[74], but the atypical DAT inhibitors possess a different binding pose, and are not correlated with any stimulant or rewarding effect in rodents[75–77] and humans[78]. Whether the distinct pharmacological profile for VLZ is due to its allosteric binding in SERT must await further investigation. Conversely, it is plausible that blocking transport alone may be responsible for beneficial as well

as adverse outcomes of the SSRIs and that allosteric inhibition may provide no clinical or adherence benefit relative to orthosteric inhibitors.

Taken together, we show that the therapeutic drug VLZ binds with high affinity to the allosteric site of SERT and that it may be able to bind to the site in two distinct poses with equal affinities. We also show that the allosteric site is larger than previously anticipated, by defining a funnel that is wedged between TM10 and 11 and capped by Tyr579 in TM12 which extends the existing site. The allosteric potency for VLZ lies in the low nanomolar range, and hence, within a therapeutic dose regimen. VLZ interacts with polar, non-polar, hydrophobic, acidic, and basic residues at the allosteric site. The elucidated molecular interactions and high-affinity binding opens for the possibility of developing improved drugs with distinct pharmacodynamic profiles, which may translate into more beneficial therapeutic actions. The allosteric site of SERT and other NSSs has likely been underexplored in structure-based drug design and we anticipate

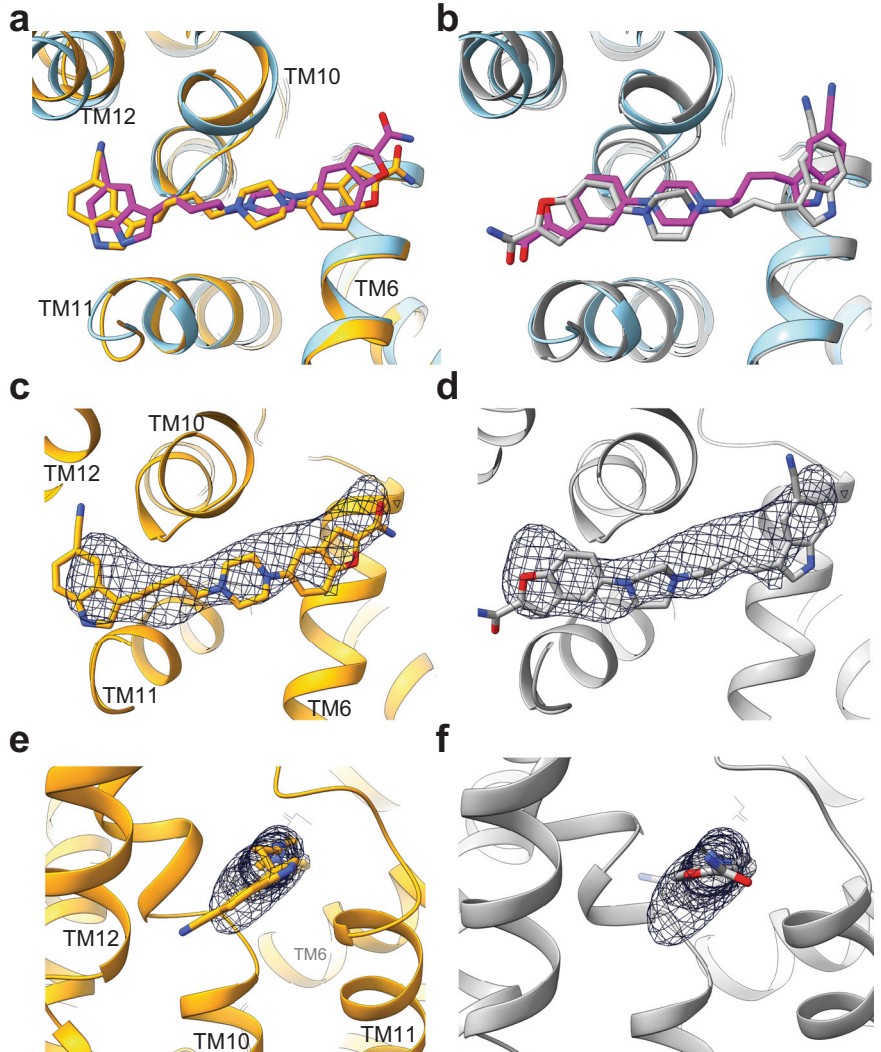

**Fig. 5 MD simulations of possible VLZ binding poses.** Zoomed view of VLZ modeled into the allosteric binding site of SERT. Parts of SERT TMs and bound IMI are removed for clarity. **a** The refined cryo-EM SERT structure (blue ribbons) with VLZ (magenta) superimposed onto the final MD simulated SERT structure (orange ribbons) with VLZ (orange) in the hypothezised pose. **b** Structure of the refined SERT:VLZ complex modeled with the 180° flipped VLZ in its binding site and superimposed onto the final MD simulated SERT structure (gray ribbons) with VLZ (gray) in the flipped pose. The MD simulations-derived structures of (**c**) SERT with VLZ bound at the hypothesized pose and (**d**) the flipped pose, both superimposed on the VLZ electron density from the cryo-EM structure. **e**, **f** As in c and d, viewed from the plane of the membrane, showing that the indole ring in the hypothesized pose could explain the density feature in the cryo-EM structure. TM: Transmembrane.

that further elucidation of the molecular mechanisms behind the allosterism in SERT could open the possibility for the discovery of drugs with activity modulating properties, such as use-dependency or increased transport rate.

## Methods

**Site-directed mutagenesis.** Human SERT was cloned into the pUbi1z vector using the NotI and XbaI. Mutations herein were generated using the two-step PCR method or, for R104K, F335L, E493N, E494Q, Y495A, F556A, P561G, and Y579A, ordered at GeneArt, Thermo Fisher (Waltham, MA). All mutations were confirmed by DNA sequencing. For mutants generated by PCR, the sense primers were (antisense primers were complementary): Y95F: cagtgattggctttgcagtggacctgggc; I172M: gcatcattgcctttacatggcttcctactacaac; Q332N: gatgcagccgctaacatcttcttctctc; S438T: caagcctgcaaacgttgtgtccaagcc.

**Membrane preparation.** COS-7 cells were transiently transfected with SERT WT or mutants using the Lipo2000 transfection protocol (Invitrogen): 2.6 μg SERT plasmid and 7.6 μL Lipofectamine were mixed each with 200 μL Opti-Mem$^R$(1X) and incubated for 20 minutes for complex formation. The mixture was added to 10 ml DMEM 1885 medium (in house) containing 7.2 million COS-7 cells and seeded in a 150 cm$^2$ flask. After 5 h, 20 ml DMEM1885 with Penicillin,

Streptomycin, and L-glutamine was added. After 72 h, the cells were harvested for membranes by adding 10 mL PBS + 5 mM EDTA, washed with 2 mL sucrose buffer (SB: 0.3 M sucrose, 120 mM NaCl, 5 mM KCl, 1.2 mM MgSO$_4$, 1.2 mM CaCl$_2$, 25 mM HEPES, pH 7.4), and lysed with one ultrasonic burst (Branson Sonifier with microtip) in 1 mL SB. Membranes were pelleted at 4500xG for 20 min, resuspended in 1 mL SB, and stored at −20 °C.

**[$^3$H]S-CIT equilibrium binding experiments.** COS-7 cells were transiently transfected as for membrane preparations, with 0.018 μg SERT plasmid, 0.055 μL Lipofectamine per 50,000 cells. 0.3 ml (containing 25.000 cells) were added to each well in 24-well plates coated with poly-ornithine. After 5 h, 0.6 ml of DMEM1885 with antibiotic was added to each well, and incubated for 2 days at 37 °C, 10% CO$_2$, and 100% humidity. The binding assays were carried in Binding Buffer (BB) (25 mM HEPES, 130 mM NaCl, 5.4 mM KCl, 1.2 mM CaCl$_2$, 1.2 mM MgSO$_4$, 1 mM L ascorbic acid, 5 mM D glucose, pH 7.4). Prior to the experiment, the cells were washed once in 400 μl of BB and the non-labeled compound (S-CIT or VLZ) was added to the cells in the indicated concentrations in a total volume of 450 μl. The assay was initiated by the addition of 50 uL 3 nM [$^3$H]S-CIT (81 Ci/mmol). Non-specific binding was determined with 1 μM paroxetine (Sigma-Aldrich). After 60 min of incubation at room temperature, the cells were washed twice with 500 μl of ice-cold BB, lysed in 250 μl 1% SDS and left for >60 min at 37 °C. All samples were transferred to 24-well counting plates (Perkin Elmer, Waltham, MA), 500 μl (of Opti-phase Hi Safe 3 scintillation fluid (Perkin Elmer) was added followed by

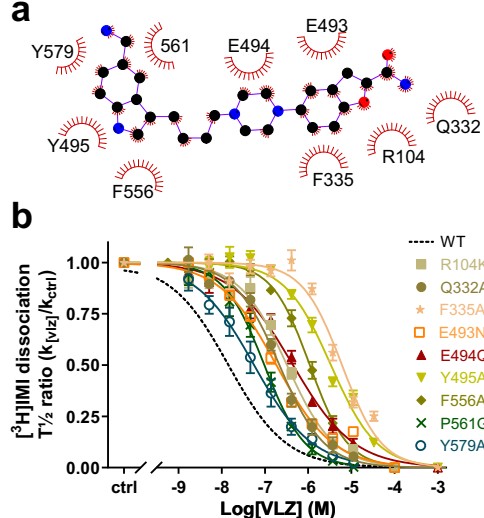

**Fig. 6 Effect of SERT S2 mutants on VLZ allosteric potency. a** Zoomed interactions between VLZ and SERT. Schematics are generated by LIGPLOT + 1.4. Each eyelash motif indicates a hydrophobic contact. **b** Effect of mutating residues in the S2 site on allosteric potency for VLZ. Point mutation of residues, shown in the cryo-EM structure to interact with VLZ, all significantly ($P < 0.001$, one-way ANOVA with Tukey's multiple comparisons test) decreases its allosteric potency to various degrees relative to SERT WT (dotted line, data shown in Fig. 2b). See allosteric potencies for SERT WT and mutants in Table 1. Experiments are performed on membrane preparations from COS-7 cells transiently expressing the indicated SERT mutants. Data are shown as means ± S.E. (error bars), $n = 3$–10. Wild-type is shown with a black dotted line. Allosteric site mutants are shown using colored symbols and solid lines defined in the legend. See Table 1 for quantitative data including allosteric potency and n-value for the individual experiments. Source data are provided as a Source Data file.

counting of the plates in a Wallac Tri-Lux β-scintillation counter (Perkin Elmer). All experiments were carried out using 10 different ligand concentrations and performed in triplicate.

**[3H]5-HT uptake**. Uptake experiments were performed at RT using 5-[1,2-3H] hydroxytryptamine ([3H]5-HT, 43,1 Ci/mmol, Perkin Elmer) on COS-7 cells transfected and seeded as described for [3H]S-CIT equilibrium binding experiments. The seeded cell number was adjusted to achieve an uptake level of maximally 10% of the total added [3H]5-HT. The uptake assays were carried out 2 days after transfection. Just prior to the experiment, the cells were washed once in 400 μL BB at room temperature. 50 μL of the tested inhibitors were added to cells in the indicated concentrations, 30 min prior to the addition of 50 μL 8–12 nM [3H]5-HT. After 3 min of incubation, the uptake reaction was stopped by washing twice with 500 μL ice-cold BB. For saturation uptake experiments, the indicated concentrations of [3H]5-HT were added and incubated for either 3 (1 and 1.9 nM), 6 (3, 3.8 and 6 nM), or 10 (7.5, 10, and 15 nM) min. Cells were lysed and transferred to counting plates as described above. Non-specific uptake was determined in the presence of 1 μM paroxetine. All determinations were performed in triplicate.

**[3H]IMI and [3H]S-CIT dissociation rate assay**. [3H]IMI (81 Ci/mmol) or [3H] S-CIT (81 Ci/mmol) were added to a membrane suspension of 1 mL transfected membranes and 3 mL membrane buffer (MB, 120 mM NaCl, 5 mM KCl, 1.2 mM MgSO4, 1.2 mM CaCl2, 25 mM HEPES, pH 7.4) to a concentration for [3H]IMI and [3H]S-CIT on 0.95 nM and 2.8 nM, respectively. Binding was incubated for 30 minutes at room temperature to reach equilibrium. Dissociation was initiated by 12x dilution of the membrane suspension with MB containing 1 μM paroxetine and the indicated concentrations of allosteric inhibitor (VLZ or S-CIT). Note: 1 μM paroxetine has no allosteric effect on the dissociation of [3H]IMI and [3H]S-CIT. The dissociation was stopped at seven time points (5–10–15–20–30–50–70 min) by rapid filtration of the samples through GF/B filters using a Tomtec cell harvester and washed for 20 s with ice-cold 20 mM HEPES, 0.2 M NaCl, pH 7.4. Non-specific binding was determined by incubating 50 μL membrane suspension in 600 μL MB with 1 μM paroxetine at 37 °C for 1 h. Experiments were performed in a water bath at a temperature where $t\frac{1}{2}$ for control dissociation (i.e., dissociation without

allosteric inhibitor) were set to approximately 15 min. All data are collected from at least three independent experiments. The allosteric potency was calculated from dissociation rate constants ($k_{[VLZ]}$) for [3H]IMI or [3H]S-CIT at different VLZ concentrations and expressed relative to the dissociation rate constant without VLZ ($k_{buf}$). IC50 values were calculated from concentration–effect curves of normalized dissociation ratio ($k_{[VLZ]}/k_{buf}$) versus log[drug] and are shown as mean values calculated from means of pIC50 and the SE interval from the pIC50 ± S.E. All data were analyzed using Prism 9 (GraphPad Software Inc., San Diego, CA).

**SERT expression and purification**. The human SERT construct used for the cryo-EM studies was the N- and C-terminally truncated WT transporter (ΔN72, ΔC13)[44,79]. Cells were solubilized in 20 mM Tris-HCl, pH 8, 100 mM NaCl containing 20 mM DDM, 2.5 mM CHS in the presence of 10 μM VLZ and 10 μM IMI and were then purified into buffer A containing 20 mM Tris-HCl, pH 8.0, 100 mM NaCl, 1 mM DDM, 0.2 mM CHS, 10 μM VLZ, and 10 μM IMI by Strep-Tactin affinity chromatography. The N- and C-termini containing GFP and purification tags were removed by thrombin digestion. SERT was mixed with 15B8 Fab at a 1:1.2 molar ratio. The resulting complexes were further purified by size-exclusion chromatography into buffer A. The peak fraction containing the SERT-15B8 Fab was concentrated to 4 mg/ml and then 100 μM VLZ and 100 μM IMI were added before cryo-EM grid preparation.

**Cryo-EM sample preparation and data acquisition**. For cryo-EM samples, 2.5 μl purified SERT-15B8 Fab complex was applied to glow-discharged Quantifoil holey carbon grids (gold, 2.0/2.0 μm size/hole space, 200 mesh). 100 μM fluorinated *n*-octyl-β-d-maltoside (final concentration) was added to the sample before freezing. After applying protein, the grids were blotted for 2 s at 100% humidity at 4 °C and plunge frozen in liquid ethane cooled by liquid nitrogen using a Vitrobot Mark IV system. Cryo-EM data were collected on a Titan Krios electron microscope equipped with a K3 direct electron detector, a BioQuantum energy filter, and operating at 300 kV. A total of 5,228 micrographs were automatically collected with SerialEM[80] at a nominal magnification of 77,160x in super-resolution counting mode with a binned pixel size of 0.648 Å/pixel. The typical defocus values ranged from −0.6 μm to −2.2 μm. The total dose was 43 e−/Å² for each stack.

**Cryo-EM data processing**. Drift correction of micrographs was performed using MotionCor2[81] and the defocus values were estimated with Gctf[82]. A total of 1,146,802 particles picked using Dog-Picker (https://github.com/craigyk/emtools)[83] were subjected to reference-free 2D classification followed by heterogenous refinement in cryoSPARC v3.2[84]. Homogeneous refinement local contrast transfer function (CTF) refinement and then non-uniform refinement was performed in cryoSPARC after recentering particles[85] (Supplementary Fig. 4). The maximum fit resolution for local CTF refinement was 3.6 Å. The 185,019 selected particles yielded a reconstruction at 3.65 Å. The resolution was estimated with the gold-standard Fourier shell correlation (FSC) 0.143 criterion[85] in cryoSPARC. The local resolution was also calculated in cryoSPARC.

**Model building and refinement**. A previous cryo-EM structure of the ts2-active SERT in complex with 15B8 Fab and 8B6 ScFv bound to ibogaine (6DZY)[20] was used as initial model; the 8B6 ScFv was removed before docking the PDB into the sharpened map in ChimeraX v0.9[86]. Manual adjustment was then performed in Coot v0.8.9.1[87] and VLZ and IMI were placed into the electron densities in Coot to generate a model. Model refinement of the coordinates was carried out in PHENIX v1.15.2-3472[88] using the real space refinement package. This iterative refinement process was repeated until the model reached optimal stereochemistry and geo-metric statistics as evaluated by MolProbity[89]. For cross-validation, the FSC curve between the refined model and half maps was calculated and compared to avoid overfitting.

**Atomistic MD simulations**. The SERT:IMI:VLZ complexes were prepared using the Maestro software tool (Schrödinger Release 2021-2: Schrödinger, LLC, New York, NY, 2021); antibody fragments were removed, and missing hydrogens atoms and side chains were added. Amino acid $pK_a$ was calculated using Epik[90]. Two Na+ ions and one Cl− ion were modeled on the basis of PDB ID: 5I71 and 5I6X[44] by protein backbone superposition of the cryo-EM structures to the X-ray crystal-lography protein structure and deleting the latter ones. Final models were pro-cessed through the Orientations of Proteins in Membrane (OPM) tool[91]. The force field parameters of protonated VLZ and IMI were parameterized according to the CHARMM General Force Field (CGenFF)[92]. The insertion of the OPM-protein complexes into a bilayer was performed with CHARMM-GUI[93]. Each hSERT:I-MI:VLZ complex was inserted in a lipid bilayer consisting of 249 1-palmitoyl-2-oleoyl-sn-glycero-3-phosphocholine (POPC) and 83 cholesterol molecules, fol-lowed by hydration and NaCl (150 mM). Approximate box dimensions: 110 × 110 x 115 Å³ (~128000 atoms). All simulations were performed with GROMACS version 2020.3 using CHARMM36m force fields[94] for SERT, CHARMM36 force fields[95] for lipids, and the TIP3P model[96] for water. To maintain the temperature, a Nosé-Hoover temperature coupling method[97] with a tau-t of 1 ps was used, and for pressure coupling, a semi-isotropic Parrinello — Rahman method[98] with a tau-p of 5 ps and a compressibility of $4.5 \times 10^{-5}$ bar$^{-1}$ was used. The temperature was

maintained at 310 K and pressure at 1 bar. Non-bonded interactions were calculated in a pairwise manner within the 12 Å cutoff, with a switching function applied between 10–12 Å. Long-range non-bonded interactions were calculated with the particle mesh Ewald (PME) method[99]. The LINCS[100] method was applied to hydrogen bonds. Periodic boundary conditions were used. 5000 steps of steepest descent minimization were performed. Next, systems were equilibrated with two sets of NPT and NVT of simulations to smoothly relax the system with overall duration of 500 ps and 6 ns respectively, during which lipids, Cα atoms, protein side chains, ligand heavy atoms and the bound ions were restrained individually by harmonic potentials with decreasing force constants (from 4 kcal/mol/Å$^2$ to 0 kcal/mol/Å$^2$) to allow for relaxation of protein side chains and hydration of the protein. The equilibrated structures were subjected to 200 ns NPT MD simulations, integrated into 2-fs time steps, and trajectories recorded every 10 ps. VMD[101] was used for visualization. The RMSD was calculated using GROMACS version 2020.3 software tools.

**MMPBSA calculation.** The MMPBSA (Molecular Mechanics Poisson–Boltzmann Surface Area) method was used to calculate the protein-ligand binding free energy of each VLZ pose. For each system, in total, the final 750 frames were extracted from the trajectories and the free energy of binding was determined using the g_mmpbsa[102] tool developed for GROMACS.

**Reporting summary.** Further information on research design is available in the Nature Research Reporting Summary linked to this article.

## Data availability
Data supporting the findings described in this manuscript are available from the corresponding authors upon request. The source data underlying Figs. 1, 2, 6, Table 1, Supplementary Figs. 2 and 3, Supplementary Table 1 and 2 are provided as a Source Data file. The data for the SERT cryo-EM structure are deposited in the Electron Microscopy Data Bank (EMDB) with accession code EMD-23545. An atomic model of SERT complexed with imipramine, vilazodone and 15B8 Fab are deposited in the wwPDB with accession code 7LWD [https://doi.org/10.2210/pdb7LWD/pdb]. Source data are provided with this paper.

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

## Acknowledgements
The authors thank Lone Rosenquist for excellent technical assistance, Vivek Kumar for discussion on vilazodone and to Prof. Antonios Kolocouris (U. Athens) for providing computational time. A portion of this research was supported by NIH grant U24GM129547 and performed at the Pacific Northwest Cryo-EM Center at OHSU and accessed through EMSL (grid.436923.9), a DOE Office of Science User Facility sponsored by the Office of Biological and Environmental Research. The work was supported in part by The Independent Research Fund Denmark (7016-00272 A to C.J.L.), The Lundbeck Foundation (R344-2020-1020 to C.J.L.), The Novo Nordic Foundation (NNF19OC0058496 to C.J.L.), The Carlsberg Foundation (CF19-0527 and CF20-0345 to C.J.L.), the NIDA Intramural Research Program (Z1A DA000610 to A.H.N), the NIH grant 5R37MH070039 (D.Y., J.C., and E.G) and the H2020 Marie Sklodowska-Curie Innovative Training Networks (860954 to I.E.K). E.G. is a Howard Hughes Medical Institute Investigator.

## Author contributions
C.J.L. conceptualized the ideas together with A.H.N. C.J.L., J.A.C., E.G., P.P. and D.Y. designed the experiments. P.P., D.Y., K.S., L.L. and J.A.C. performed the experiments and data analysis together with C.J.L. MD simulations were performed and analyzed by I.E.K. All authors were involved in data interpretation. C.J.L., D.Y. and J.A.C. prepared the manuscript with significant contribution from all authors.

## Competing interests
The authors declare no competing interests.
