## [Peer Review File · Nature Communications]

The antidepressant drug vilazodone is an allosteric inhibitor of the serotonin transporterREVIEWER COMMENTS

Reviewer #1 (Remarks to the Author):

In this contribution by Plenge et al., a complex of SERT transporter with the drugs vilazodone and imipramine obtained by cryo-EM is reported. The structural analysis is fortified by functional analysis and in general the work is solid and conclusions are supported by the experimental evidence. The main novelty is that vilazodone binds not to the main binding site, but rather to the allosteric site, which can be used for further drug development.

My main concern is a relatively low resolution of 3.8 Å, which the authors for some reason are trying to 'sell' as the high resolution (lines 256, 275). Even worse, the pdb validation report shows the estimated resolution of ~4.8 Å (at FSC of 0.143), so the authors should explain this discrepancy. That is actually reflected in rather poor densities for ligands (Fig. 3 panel b and c), hence I recommend the authors to use docking scoring to assist placing ligands in the poses with the lowest energy.

Minor issues:

line 60 - the word 'isomerize' is not very suitable, rather use 'rearrange'

line 211 - please order the residues with the ascending numbering

line 213 - no space between 'interactions' and 'between'

line 337 - 339 - CO₂ and CaCl₂ - use subscript

line 352 - COS-7 transfected, I guess the word 'cells' is missing here

line 356 - what is BB buffer?

line 369 - the word 'obtain' is misspelled

line 375 - no space '20mM'

line 396 - why was a fluorinated detergent used? if there was a problem with orientations of particles, please elaborate

Fig. 2 enlarge the inset (illegible)

Fig. 4 please avoid combination of red and green

Reviewer #2 (Remarks to the Author):

In the well-written manuscript "The antidepressant drug vilazodone is an allosteric inhibitor of the serotonin transporter" Plenge and colleagues use pharmacological, site-directed mutagenesis, and high-resolution cryo-EM strategies to characterize vilazodone interaction with the human serotonin transporter. This research group is known for quality work and likewise these studies are high caliber with outcomes that are of primary importance to the areas of antidepressants and monoamine neurotransmitter transporters and is relevant to multiple research areas. Statistical analyses are appropriately applied and valid.

Impact: The findings of this study represent a significant step forward in understanding antidepressants that exhibit novel non-orthosteric binding modalities while allosterically inhibiting serotonin transporter function. This foundational work could lead to compounds with unique modulatory characteristics that enhance clinical outcomes. In addition the work identifies the SERT S1 binding site for the tricyclic antidepressant imipramine.

Comment and Concerns:

1. Use of "expanding the extracellular vestibule" in the abstract seems inaccurate as the outer vestibule is not actually expanded, rather it is just the identification of additional binding domains within the existing extracellular vestibule thereby expanding the boundaries of S2.
2. There are several instances in the manuscript that discuss antidepressant side effects. The impact of modulators that are "solely allosteric" on common side effects should include discussion that it is possible that transport blockade alone may be responsible for adverse outcomes and that allosteric inhibition may provide no clinical or adherence benefits especially in light of the estimates of the percent occupancy of SERT that must occur before clinical benefit.
3. Page 3, it is stated that the "empty transporter" rectifies. However, has it been determined that the

transporter is void of ions such as K and Na when it rectifies?

4. This is just a thought question which I pondered while reading the manuscript. Could the high affinity of VLZ identified in the competitive assays in the manuscript be dependent on occupancy of S1 by another ligand? Could VLZ exhibit higher or lower affinity to SERT if an S1-binding compound was not co-administered?
5. The relatively steep VLZ competitive inhibition slope is intriguing (Hill slope >1). Is it possible that VLZ binding is slow perhaps through an induced fit process that is not compensated for in your pre-incubation time and therefore the binding assay is not occurring under equilibrium conditions?
6. On Page 6 the sentence "Using the same experimental setup as in Fig. 1a, we found that all three mutants are capable of transporting 5-HT with a KM similar or slightly higher than observed for SERT WT (Extended Data Fig. 2)." This was confusing as extended figure 2 does not show KM values. Should this refer to Extended data table 1?
7. On page 6 when discussing the residues involved in the high affinity S1 site consider referencing Barker et al., 1998, Larsen et al., 2004, and Henry et al., 2006.
8. On page 7 the authors state "The high allosteric potency of VLZ suggests that it binds to the S2 site and likely slows dissociation by occlusion of the exit pathway from the S1 site." This seems slightly overstated as VLZ could also bind outside of S2 and stabilize an occluded or alternate SERT conformation that inhibits radioligand release from S1.
9. Page 8. Suggest changing the sentence "The heteroaryl piperazine moiety of VLZ overlaps with S-CIT and Lu AF60097 (Extended Data Fig. 1)," to "The heteroaryl piperazine moiety of VLZ overlaps with S2-bound S-CIT and Lu AF60097 (Extended Data Fig. 1)," for clarity.
10. On Page 10, the authors posit that residues chosen for substitution define the VLZ binding site due to the correlation with mutation and loss of allosteric activity. It would have been more convincing if the allosteric properties of S-CIT or IMI were evaluated with these mutants as one might expect some residues to impact all three ligands whereas others would exhibit more profound effects on VLZ pharmacology. Perhaps you performed some of these analyses?
11. On page 12 the authors state "If one site is occupied by 5-HT, IMI or disrupted by mutations, it will simply only associate with the other site." However, this idea requires more mechanistic explanation in regards to how this would occur under equilibrium conditions if VLZ were added prior to the radiolabeled ligand.
12. Change "affinity" to "potency" in the title of Extended Data Figure 2.
13. On Page 11 in the sentence (iv) we provide high-resolution structural evidence for the association of VLZ to the S2 site (Fig 3,4). Please remove "we provide" as it is unnecessary and makes the sentence unbalanced.
14. There was a recent manuscript detailing VLZ ligand docking and MD simulation to SERT which may be of interest although their findings suggest VLZ binds primarily is S1 and extends into S2. Zhang Y, Zheng G, Fu T, Hong J, Li F, Yao X, Xue W, Zhu F. The binding mode of vilazodone in the human serotonin transporter elucidated by ligand docking and molecular dynamics simulations. *Phys Chem Chem Phys*. 2020 Mar 7;22(9):5132–5144. PMID: 32073004

Review by Keith Henry

Reviewer #3 (Remarks to the Author):

This manuscript describes the interaction of vilazodone (VLZ) with the serotonin transporter SERT. This is important because SERT is the site of action of antidepressants which competitively inhibit this transporter. These so-called selective serotonin reuptake inhibitors (SSRIs) are in fact not so selective. Clinically approved VLZ is a new SSRI with less side-effects than its "classical" counterparts and therefore the Authors set out to characterize the VLZ/SERT interaction with mechanistic and structural approaches. The latter is achieved by obtaining a novel SERT/VLZ/IMI structure. They find that in contrast to the other SSRIs, which are competitive inhibitors and bind to the central S1 site, VLZ is a non-competitive inhibitor of SERT which binds to the more peripheral S2 site. This work opens up new possibilities to find even more selective antidepressants.

This is a strong story and my critique mainly deals with the presentation of the Figures documenting the structural work.

1. Lines 187-189: the structure is "outward-open" but the extracellular gate is found to be closed. This should be better explained, perhaps by adding a cartoon illustration.
2. Lines 194- 218: the presentation would benefit from adding enlarged views and perhaps also adding more viewpoints.
3. Lines 266-269: show this in an improved Fig. 4a.
4. Line 222: I cannot see Glu493 in Fig. 5A.
5. Lines 164-167: can you provide an explanation? How about docking?

Minor

Line 224: Gln332 in Fig 5a but Glu332 in text; 227: "detectable" instead of "viable"; 193 add Fig. 3b in parenthesis; 208 4b instead of 4a; 190 stimulations.

Reviewers' comments

We thank all reviewers for their insightful comments. We find that the manuscript has greatly improved after incorporating all the suggestions herein. Please see our replies inserted as point-by-point in blue below your comments.

Reviewer #1

In this contribution by Plenge et al., a complex of SERT transporter with the drugs vilazodone and imipramine obtained by cryo-EM is reported. The structural analysis is fortified by functional analysis and in general the work is solid and conclusions are supported by the experimental evidence.

We thank the reviewer for complementing our work.

Comment 1: My main concern is a relatively low resolution of 3.8 Å, which the authors for some reason are trying to 'sell' as the high resolution (lines 256, 275). Even worse, the pdb validation report shows the estimated resolution of ~4.8 Å (at FSC of 0.143), so the authors should explain this discrepancy. That is actually reflected in rather poor densities for ligands (Fig. 3 panel b and c), hence I recommend the authors to use docking scoring to assist placing ligands in the poses with the lowest energy.

Reply: We thank the reviewer for making this point. As described below, we have improved the resolution of the cryo-EM structure by extensive reprocessing and added a new results section with molecular dynamics simulations to clarify the binding poses of VLZ and IMI. The results turned out to be very interesting and has strengthened our findings.

First, we re-processed the data and we were able to improve the resolution to 3.65 Å with the local resolution around 3.0 Å (see new Supplementary Fig. 4). The map quality was significantly improved as shown in the new Fig. 3. The ligand densities, especially for VLZ were better resolved (Fig. 3c, d), which has enabled us to dock the ligands into the densities more accurately. The lower resolution estimation reported by the pdb validation is because no mask was used to calculate the FSC by the pdb server. In the revised manuscript, we have reported both the unmasked and masked FSC resolution. These values more closely correspond to the value reported by the pdb validation server.

Second, as suggested, we then performed a MD simulation and docking scoring based on the refined structure. To further ensure a correct binding pose, we added a MD simulation of the ligands rotated by 180° in their binding pockets. We found that because of the pseudosymmetric structure of VLZ it does fit quite well in this flipped pose. The MD simulations also revealed very similar RMSDs of VLZ in its hypothesized and flipped poses. A calculation of the protein-ligand binding free energy of each VLZ pose were also equally high. In all, this suggests that VLZ may bind with equally high affinities in both poses. We find this a very interesting feature, which we believe is quite uncommon for

high-affinity ligands at this size. However, the improved resolution does strongly imply that the hypothesized pose is the predominant one in our cryo-EM structure (see new Fig. 5).

The refinement of the cryo-EM structure and new series of MD simulations have substantial improvements to the manuscript. We have made extensive revisions throughout the cryo-EM results section and added a new section about the MD simulations (line 226-253). We have added a new Fig. 5, Supplementary Figures 5 + 6 and Supplementary video 1 + 2. We also have a new section in Methods (line 479-516). Figure 3 + 4, supplementary Fig. 4 and Supplementary Table 3 have been modified to illustrate the new data.

Minor issues:

Comment 1-5, 7+8

line 60 - the word 'isomerize' is not very suitable, rather use 'rearrange'

line 211 - please order the residues with the ascending numbering

line 213 - no space between 'interactions' and 'between'

line 337 - 339 - CO₂ and CaCl₂ - use subscript

line 352 - COS-7 transfected, I guess the word 'cells' is missing here

line 369 - the word 'obtain' is misspelled

line 375 - no space '20mM'

Reply: We have fixed these typos and added missing words and space as well as numerically re-ordered the residues in line 211.

Comment 6: line 356 - what is BB buffer?

Reply: BB is Binding Buffer (25 mM HEPES, 130 mM NaCl, 5.4 339 mM KCl, 1.2 mM CaCl₂, 1.2 mM MgSO₄, 1 mM L ascorbic acid, 5 mM D glucose, pH 7.4). It is indicated in line 383. We have now emphasized it by capitalizing the initial letters.

Comment 9: line 396 - why was a fluorinated detergent used? If there was a problem with orientations of particles, please elaborate

Reply: Fluorinated octylmaltoside was used to improve the particle distribution and also allows for more homogeneous ice across the grid. We have used fluorinated octylmaltoside in several other cryo-EM studies of SERT for the same purpose.

Comment 8: Fig. 2 enlarge the inset (illegible)

Reply: The inset has been inserted as a Fig 2b so it has the size of the other panels. We have revised the figure legend accordingly.

Comment 9: Fig. 4 please avoid combination of red and green

Reply: Thanks for pointing this out. The residue color has been changed from green to violet.

Reviewer #2

This research group is known for quality work and likewise these studies are high caliber with outcomes that are of primary importance to the areas of antidepressants and monoamine neurotransmitter transporters and is relevant to multiple research areas. Statistical analyses are appropriately applied and valid.

We thank the reviewer for this positive opinion on our efforts and findings.

Comment 1. Use of “expanding the extracellular vestibule” in the abstract seems inaccurate as the outer vestibule is not actually expanded, rather it is just the identification of additional binding domains within the existing extracellular vestibule thereby expanding the boundaries of S2.

Reply: We agree. The sentence now reads (line 35-37): “Our SERT structure with bound imipramine and vilazodone reveals a unique binding pocket for vilazodone, expanding the boundaries of the extracellular vestibule.”

Comment 2. There are several instances in the manuscript that discuss antidepressant side effects. The impact of modulators that are “solely allosteric” on common side effects should include discussion that it is possible that transport blockade alone may be responsible for adverse outcomes and that allosteric inhibition may provide no clinical or adherence benefits especially in light of the estimates of the percent occupancy of SERT that must occur before clinical benefit.

Reply: We agree with the reviewer that this is also a likely scenario. To increase the emphasis on this possibility, we have added the following in the Discussion (line 338-340): “*Conversely, it is plausible that blocking transport alone may be responsible for beneficial as well as adverse outcomes of the SSRIs and that allosteric inhibition may provide no clinical or adherence benefit relative to orthosteric inhibitors.*”

Comment 3. Page 3, it is stated that the “empty transporter” rectifies. However, has it been determined that the transporter is void of ions such as K and Na when it rectifies?

Reply: No. This is a bold statement. Thank you for putting this to our attention. In fact, it likely contains K⁺ (and maybe Cl⁻) in the return step. We have specified the sentence to the following (line 65-67): “*The transporter then transitions back to the outward-open conformation^{6,20,21}. For SERT and LeuT, the return step can involve a counter-transport of K⁺^{22,23}.*”

Comment 4. This is just a thought question which I pondered while reading the manuscript. Could the high affinity of VLZ identified in the competitive assays in the manuscript be dependent on occupancy of S1 by another ligand? Could

VLZ exhibit higher or lower affinity to SERT if an S1-binding compound was not co-administered?

Reply: That is indeed possible. The allosteric potency of vilazodone bound to S2 is dependent on the S1-bound ligand with an affinity of 14 nM when [³H]IMI is bound and 250 nM in the presence of [³H]S-CIT (Fig 2b). Its potency in inhibiting [³H]5-HT transport is ~1 nM (Suppl. Table 1). To know its binding affinity without an S1-binding ligand would require radiolabeled vilazodone, which is not available. The experiment would be interesting, but unfortunately not feasible at this time.

Comment 5. The relatively steep VLZ competitive inhibition slope is intriguing (Hill slope >1). Is it possible that VLZ binding is slow perhaps through an induced fit process that is not compensated for in your pre-incubation time and therefore the binding assay is not occurring under equilibrium conditions?

Reply: We doubt that this is the case. To assess equilibrium conditions, we initially, we performed VLZ binding for 1hr at 5 drgC as well as room temperature for 30 min and 1, 2, 3, 4, 5 and 6 hrs. We observed no change in affinity or in Hill slope between any of the conditions.

Comment 6. On Page 6 the sentence “Using the same experimental setup as in Fig. 1a, we found that all three mutants are capable of transporting 5-HT with a KM similar or slightly higher than observed for SERT WT (Extended Data Fig. 2).” This was confusing as extended figure 2 does not show KM values. Should this refer to Extended data table 1?

Reply: We agree. What is shown on Ext Data Fig. 2 is the IC50 values. Since the experiments are performed under the same conditions, they correlate with the K_M values. The quantitative data are shown in Ext Data Table 1. We have added the reference to Ext Data Table 1 in the sentence.

Comment 7. On page 6 when discussing the residues involved in the high affinity S1 site consider referencing Barker et al., 1998, Larsen et al., 2004, and Henry et al., 2006.

Reply: Thank you for reminding us of these milestone papers. The references have been inserted.

Comment 8. On page 7 the authors state “The high allosteric potency of VLZ suggests that it binds to the S2 site and likely slows dissociation by occlusion of the exit pathway from the S1 site.” This seems slightly overstated as VLZ could also bind outside of S2 and stabilize an occluded or alternate SERT conformation that inhibits radioligand release from S1.

Reply: We agree. We have modified the sentence to be less specific (line 173-175): “The high allosteric potency of VLZ could suggest that it binds in the extracellular vestibule and thereby slows dissociation by occlusion of the exit

pathway from the S1 site.”

Comment 9. Page 8. Suggest changing the sentence “The heteroaryl piperazine moiety of VLZ overlaps with S-CIT and Lu AF60097 (Extended Data Fig. 1), “ to “The heteroaryl piperazine moiety of VLZ overlaps with S2-bound S-CIT and Lu AF60097 (Extended Data Fig. 1),” for clarity.

Reply: Corrected.

Comment 10. On Page 10, the authors posit that residues chosen for substitution define the VLZ binding site due to the correlation with mutation and loss of allosteric activity. It would have been more convincing if the allosteric properties of S-CIT or IMI were evaluated with these mutants as one might expect some residues to impact all three ligands whereas others would exhibit more profound effects on VLZ pharmacology. Perhaps you performed some of these analyses?

Reply: The purpose of the systematic mutagenesis of the interacting residues was not to verify the VLZ binding site – we find the cryo-EM structure quite convincing in that aspect – but to investigate the contribution of each side chain residue to the VLZ binding affinity. We acknowledge that the purpose was not entirely clear. We have rephrased the sentence to (line 259-261):

“To further investigate the involvement of each side chain, we mutated each residue, one-by-one, and monitored the impact on the allosteric potency of VLZ (Fig 6b, Table 1)”.

- and further in line 273-276: “The fold change in allosteric potency is not only dependent on the interaction of the residue and VLZ, but also on the ability of the substituted residue to compensate for the WT environment. The isolated contribution of each residue might deviate somewhat from the observed affinity changes.”

We have investigated the allosteric binding properties of S-CIT in an earlier study (Plenge et al. 2012 JBC). Indeed, we found a partial overlap between interacting residues. IMI has, if any, very weak allosteric properties (Plenge et al 1990 Eur J Pharmacol). We doubt that it would provide meaningful results to characterize its allosteric binding properties with mutagenesis studies.

Comment 11. On page 12 the authors state “If one site is occupied by 5-HT, IMI or disrupted by mutations, it will simply only associate with the other site.” However, this idea requires more mechanistic explanation in regards to how this would occur under equilibrium conditions if VLZ were added prior to the radiolabeled ligand.

Reply: We agree that this was very speculative. We have therefore deleted the sentence.

Comment 12. Change “affinity” to “potency” in the title of Extended Data Figure 2.

Reply: Done.

Comment 13. On Page 11 in the sentence (iv) we provide high-resolution structural evidence for the association of VLZ to the S2 site (Fig 3,4). Please remove “we provide” as it is unnecessary and makes the sentence unbalanced.

Reply: It has been removed. Thanks.

Comment 14. There was a recent manuscript detailing VLZ ligand docking and MD simulation to SERT which may be of interest although their findings suggest VLZ binds primarily is S1 and extends into S2. Zhang Y, Zheng G, Fu T, Hong J, Li F, Yao X, Xue W, Zhu F. The binding mode of vilazodone in the human serotonin transporter elucidated by ligand docking and molecular dynamics simulations. Phys Chem Chem Phys. 2020 Mar 7;22(9):5132–5144. PMID: 32073004

Reply: We thank the reviewer for pointing this out. The paper is currently reference #53 in our manuscript. We describe their findings in the introduction (line 110-115). We relate our S1 mutagenesis results to their MD simulation in the Results section (line 142-153) and in Discussion (line 315-322)

Reviewer #3

Comment 1: Lines 187-189: the structure is "outward-open" but the extracellular gate is found to be closed. This should be better explained, perhaps by adding a cartoon illustration.

Reply: We agree that this is in discrepancy to observations from LeuT where the outward-open conformation is largely defined by the breakage of the salt bridge. It is entirely possible that SERT possess yet an outward-open conformation where the salt bridge is broken. Indeed, a recent elaborate MD simulation have shown an outward-open SERT Apo form with a broken salt bridge (Chan, Selvam, Young, Procko, Shukla (2019): ChemRxiv. <https://doi.org/10.26434/chemrxiv.9922301.v2>). However, the structure is similar to previously solved SERT structures with access to the S1 site from the extracellular site, but with an intact salt bridge. We have therefore decided to use the same nomenclature for this structure.

We have now displayed the extracellular pathway in a slice view in Figure 3b which clearly shows that it is the vilazodone, which blocks the pathway to the central binding site.

To further elaborate on this, we have also added the following paragraph (line 323-329): “The solved SERT structure is similar to previously solved structures with antidepressants bound in the S1 site^{44,70,71} though the Arg104-Asp493 salt bridge is intact the S1-bound ligands are solvent accessible to the extracellular

environment. Accordingly, the conformation is outward open. This is in contrast to the solved LeuT structures where an intact salt bridge excludes solvent to the S1 site and is thus an outward-occluded conformation. It is indeed possible that the SERT transport cycle also includes an outward-open conformation with a broken salt bridge.”

Comment 2: Lines 194- 218: the presentation would benefit from adding enlarged views and perhaps also adding more viewpoints.

Reply: We agree that it would benefit from more details. We have enlarged the views for Fig. 3 and Fig. 4. To increase the perspectives in the relative binding sites we have inserted a new Fig 3b showing a new slice view of SERT illustrating the interrelationship between IMI and VLZ binding. We have also added one more panel in Fig. 3f. providing one more view of the allosteric site bound drugs.

Comment 3: Lines 266-269: show this in an improved Fig. 4a.

Reply: The comparison of Lu AF60097, S-CIT and VLZ is shown in the new improved Fig. 3f. A reference to the figure was missing in the mentioned lines. This have now been inserted.

Comment 4: Line 222: I cannot see Glu493 in Fig. 5A.

Reply: Thank you for pointing this out. We have revised the now Fig. 6a and included Glu493.

Comment 5: Lines 164-167: can you provide an explanation? How about docking?

Reply: We have recently provided a detailed explanation for the differences in allosteric potency for citalopram when either [³H]S-CIT or [³H]IMI is bound in the S1 site. In that paper, we provided almost 1 ms of MD simulations supported with experimental data. It would be outside the scope of this manuscript to add this kind of elaborate data. However, when we compare the current structure with our previous structure with S-CIT in the S1 site (Coleman et al. 2016 Nature), we see a difference in the conformation of Phe341 and Phe335. This could likely cause the difference. We do, however, find it too speculative to add it into the text.

Comment 6:

*Line 224: Gln332 in Fig 5a but Glu332 in text;

*227: "detectable" instead of "viable";

Reply: The typos have been corrected.

Comment 7: 193 add Fig. 3b in parenthesis; 208 4b instead of 4a;

Reply: Thanks, figure citations have been revised.

Comment 8: 190 stimulations.

Reply: MD simulation is a method. It should be “simulation” instead of stimulation.

REVIEWER COMMENTS

Reviewer #1 (Remarks to the Author):

I am glad to see that the authors appreciated the given feedback and performed additional steps (docking, MD, data reprocessing) to improve the manuscript and I believe they did it pretty well. The revised manuscript is nice and solid piece of work. I recommend it for publication in Nature Communications.

A few tiny typos to correct:

line 257 - 'Is composed of residues...' I believe the word 'It' is missing - It is composed of residues...

line 455 - The total dose was 43 e-/Å² for each stack, should be e-/Å²

Reviewer #2 (Remarks to the Author):

the authors have responded to all of my questions and concerns and I recommend publishing.

Reviewers' comments

We thank all reviewers for the positive feedback and are delighted to have their acceptance.

Please see our replies inserted as point-by-point in blue below your comments.

Reviewer #1

I am glad to see that the authors appreciated the given feedback and performed additional steps (docking, MD, data reprocessing) to improve the manuscript and I believe they did it pretty well. The revised manuscript is nice and solid piece of work. I recommend it for publication in Nature Communications.

Thank you so much. We are also very satisfied with the outcome of the revisions.

A few tiny typos to correct:

line 257 - 'Is composed of residues...' I believe the word 'It' is missing - It is composed of residues...

line 455 - The total dose was 43 e-/Å² for each stack, should be e-/Å²

Both have been corrected exactly as suggested.

Reviewer #2

the authors have responded to all of my questions and concerns and I recommend publishing.

Thank you so much for the positive feedback.